# Tomato Biostimulation with Nanochitosan–Iodine Complexes: Enhancing Antioxidant Metabolism

**DOI:** 10.3390/plants14050801

**Published:** 2025-03-05

**Authors:** Luz Leticia Rivera-Solís, Hortensia Ortega-Ortiz, Adalberto Benavides-Mendoza, María Liliana Flores-López, Armando Robledo-Olivo, Susana González-Morales

**Affiliations:** 1Program in Protected Agriculture, Universidad Autónoma Agraria Antonio Narro, Saltillo 25315, Mexico; luzletirs@gmail.com; 2Department of Advanced Materials, Centro de Investigación en Química Aplicada, Saltillo 25294, Mexico; hortensia.ortega@ciqa.edu.mx; 3Horticulture Department, Universidad Autónoma Agraria Antonio Narro, Saltillo 25315, Mexico; abenmen@gmail.com; 4Centro de Investigación e Innovación Científica y Tecnológica, Universidad Autónoma de Coahuila, Avenida 3 y 16, Colonia Lourdes, Saltillo 25070, Mexico; lilianaflores@uadec.edu.mx; 5Department of Food Science and Technology, Universidad Autónoma Agraria Antonio Narro, Calzada Antonio Narro 1923, Saltillo 25315, Mexico; armando.robledo@uaaan.edu.mx; 6Secretariat of Science, Humanities, Technology and Innovation, Universidad Autónoma Agraria Antonio Narro, Saltillo 25315, Mexico

**Keywords:** antioxidant enzymatic activity, biostimulant, gene expression, potassium iodate, potassium iodide, total phenols

## Abstract

Biostimulants are currently essential for agriculture as they increase crop productivity and quality sustainably. The aim of this work was to evaluate the effects of biostimulation on the application of nanochitosan–iodine complexes (nCS-I) on tomato plants. Leaf samples were taken for analysis of total protein content, photosynthetic pigments, antioxidant enzymatic activity, mineral and iodine contents, gene expression, and shelf life in tomato fruit. The catalase (CAT), glutathione peroxidase (GPX), ascorbate peroxidase (APX), and superoxide dismutase (SOD) activities increased significantly with the application of nanochitosan (nCS) and nanochitosan–potassium iodate (nCS-KIO_3_) and nanochitosan–potassium iodide (nCS-KI) complexes and the iodine salts potassium iodate (KIO_3_) and potassium iodide (KI). The total protein content and photosynthetic pigments also increased significantly with the application of the treatments. The mineral and iodine contents did not change with the application of the treatments. Similarly, overexpression of the *SOD*, *GPX*, and *CAT* genes was observed. Finally, in the shelf life test, an increase in the total phenols and antioxidant capacity was observed with the application of the treatments. This study shows that the use of nCS-I complexes can modulate different transcriptional and post-translational processes with possible synergistic effects on the antioxidant metabolism of tomato plants.

## 1. Introduction

In recent years, in the quest to sustainably feed the growing world population, crop production has been limited to the adaptability of plants to climate change and tolerance to different types of abiotic and biotic stress factors such as high light intensity, adverse temperature, drought, or soil salinity and pathogen and herbivore colonization [1]. Consequently, the natural resilience system of plants must cope with the overproduction of reactive oxygen species (ROS) such as peroxide, superoxide, and hydroxyl and singlet oxygen, among others [2]. ROS are naturally produced in plants as a result of cellular metabolism and play an important role as messengers in cell signaling to regulate plant growth and development, but exceeding a concentration threshold causes oxidative stress [3]. Plants must maintain a balance between the generation and elimination of ROS to avoid protein denaturation, lipid peroxidation, and nucleotide degradation, which results in cellular damage or cell death [4]. Under stress circumstances, plants modify physiological and biochemical parameters by initiating signaling that can directly modify enzymatic activities or the signaling cascade finally initiates the regulation of gene expression [5]. In this sense, plants are equipped with a complex antioxidant system consisting of enzymatic and non-enzymatic components that scavenge or inhibit the oxidative action of ROS to prevent or delay cellular damage [6]. Enzymatic antioxidant defense plays a major role in protection against oxidative stress and comprises superoxide dismutase (SOD), catalase (CAT), ascorbate peroxidase (APX), peroxidase (POX), monodehydroascorbate reductase (MDHAR), dehydroascorbate reductase (DHAR), glutathione *S*-transferase (GST), glutathione peroxidase (GPX), alternative oxidase (AOX), and peroxiredoxin (Prx), while ascorbic acid (AsA), glutathione (GSH)**,** carotenoids, flavonoids, and α-tocopherol comprise the non-enzymatic antioxidative defense line [7].

In this scenario, alternatives have been sought to promote the endogenous mechanism of plants that allow for the mitigation of the adverse effects of oxidative stress and a reduction in the use of chemical inputs in agriculture [8]. Plant biostimulation is an applied technique that consists of a biological process in response to environmental factors, physical, chemical, and biological, and causes the adaptive modification of metabolic processes, in such a way that the plant makes adjustments that lead to greater efficient use of environmental resources and stress mitigation [9]. Currently, the use of biostimulants promotes natural plant processes that benefit the absorption of nutrients, increase tolerance to biotic or abiotic stress, increase yield, and improve the nutritional quality of fruits [10]. Biostimulants are defined as any substance or microorganism applied to plants for the purpose of improving their nutritional efficiency, stress tolerance, or quality characteristics, regardless of their nutrient content [11]. In the classification of biostimulants, biopolymers such as chitosan (CS) are considered some of the most abundant natural biopolymers; it is a polycationic polymer that is formed when chitin is deacetylated; this polymer is found in shrimp, crustaceans, insects, fungi, mollusks, and other marine organisms [12]. Chitosan has attracted worldwide attention for its applications in agriculture, specifically for its ability to bind other compounds, allowing for the delivery of nutrients, pesticides, and biomolecules to plant systems, and for its properties of stimulating plant growth, yield, and tolerance to biotic and abiotic stress [13]. In this sense, the application of chitosan as a biostimulant in plants strengthens the plant defense system, promotes the biosynthesis of protective biomolecules, such as phytoalexins, and regulates the expression of defense genes through the activation of the MAP-kinase pathway [14]. In addition, chitosan is involved in the synthesis of alkaloids and plant growth regulators while inhibiting the growth of pathogenic microorganisms, which contributes to the protection of plants and strengthens their response to biotic and abiotic stress [12]. Recently, chitosan nanoparticles (nCS), which are used as micronutrient nanocarriers, have been shown to be more efficient than the original material because of their higher charge surface density, larger surface area, and better cellular uptake [15]. In turn, the synthesis of nCS allows for improved binding of ions through ion exchange, physical sorption, and inter- and intramolecular trapping and the formation of more stable complexes, which improve the absorption and release of compounds by optimizing their availability and effectiveness in agricultural applications [16].

nCS can be used in plant production systems as a single compound or combined with other polymers and elements [17]. Iodine is an essential element for the life of mammals and is involved in the production of hormones by the thyroid gland, which plays critical roles in cellular metabolism and the regulation of growth and development, especially in the brain [18]. In 2004, the WHO estimated that one-third of the world’s population (2 billion people) was at risk of iodine deficiency disorders (IDDs) due to inadequate dietary iodine intake [19,20]. Therefore, the recommended daily intake of iodine is 90 µg for preschool children, 120 µg for children aged 6–12 years, 150 µg for adults over 12 years, and 200 µg for pregnant and lactating women to prevent IDDs, which cover a wide spectrum of diseases, including hypothyroidism, goiter, and adverse obstetric outcomes [21]. However, it is only considered beneficial for terrestrial plants and plays an important role in the central antioxidant system of plants, generating detoxification of ROS through activation of antioxidant defense [22]. In addition, this element regulates plant metabolism at the level of gene expression and protein activities associated with stress tolerance, redox regulation, photosynthesis machinery, and plant development [23]. Nowadays, it is well recognized that exogenous applications of iodine in the form of potassium iodide (KI) and potassium iodate (KIO_3_) have positive effects such as growth promotion, antioxidant production, and increased tolerance to abiotic stress and can be assimilated by the roots, stems, or leaves of crops such as tomato, lettuce, and melon, among others [22,24,25]. The tomato crop has emerged as an excellent candidate for agronomic biostimulation with iodine because the efficient translocation of iodine through the phloem has been demonstrated, allowing it to increase the iodine content in the fruit at concentrations that also promote the growth and general development of the plant [26]. Therefore, the objective of this research was to evaluate the biostimulation effects of nanochitosan–iodine complexes (nCS-I) in tomato plants.

## 2. Results

### 2.1. Total Protein

Figure 1 shows the results of the total protein content, revealing significant differences between treatments throughout the evaluation period. At 16 DAT, compared with T0, nCS-KIO_3_-25 increased by 46.37%, and nCS-KI-5 increased by 37.71%. Similarly, at 46 DAT, KIO_3_-25 increased by 66.45%, nCS-KIO_3_-25 increased by 54.11%, KI-25 increased by 49.42%, nCS-KIO_3_-5 increased by 43.49%, and nCS increased the total protein content by 42.40% compared with the absolute control. At 76 DAT, compared with that at T0, the application of nCS was 31.90% greater. Finally, at 106 DAT, the application of nCS-KI-5 increased by 23.64% compared with that at T0. Moreover, the amount of total proteins has a highly significant negative correlation (R ≥ 0.8) with the enzymatic activity of APX, GPX, SOD, and CAT.

### 2.2. Antioxidant Enzymatic Activity

The results of the enzymatic activity analysis indicated that there were significant differences between the treatments (Figure 2).

In terms of glutathione peroxidase (GPX) activity (Figure 2a), the application of the treatments resulted in differences from the absolute control at 16, 76, and 106 DAT. Compared with the absolute control, the nCS treatment increased the activity of GPX by 102.50% at 16 DAT. At 46 DAT, no differences were observed between the absolute control and treatments. Compared with the absolute control, the application of KIO_3_-25 and NPsCS-KI-25 resulted in increases of 39.34 and 40.98%, respectively, at 76 DAT. At the last evaluation (106 DAT), compared with the absolute control, KIO_3_-25 increased activity by 75%.

The ascorbate peroxidase (APX) activity at 76 DAT only differed from that of the absolute control (Figure 2b). The application of KI-25 increased by 49.46% with respect to the absolute control.

CAT activity (Figure 2c) with the application of nCS and nCS-KIO_3_-5 exceeded the absolute control by 77.92 and 66.38%, respectively, at 16 DAT. In the second sampling (46 DAT), all of the treatments were reduced with respect to the absolute control and KIO_3_-5. At 76 DAT, the application of KI-5 was significantly high in all of the treated plants compared with the absolute control of 200.95%. In addition, the application of nCS-KIO_3_-5 and KIO_3_-25 surpassed the absolute control by 126.59 and 118.13%, respectively. At the last sampling (106 DAT), significant differences were observed between the absolute control and the application of nCS-KIO_3_-25 (112.38%), KIO_3_-25 (93.94%), nCS-KI-25 (64.09%), or KIO_3_-5 (62.83%).

SOD activity (Figure 2d) differed from that of the absolute control. At 16 DAT, the application of nCS-KI-5 and KI-25 statistically exceeded the absolute control by 58.81 and 43.41%, respectively. In the second sampling, the application rates of the nCS, nCS-KIO3-25, KIO_3_-5, and KIO_3_-25 increased by 97.78, 78.04, 61.74, and 42.79%, respectively. At 76 and 106 DAT, no treatments increased with respect to the absolute control.

In general terms, the study of enzymatic activity is key to understanding the impact of treatment on the antioxidant metabolism of plants that have undergone a biostimulation process. In this sense, the activity of antioxidant enzymes such as SOD, CAT, APX, and GPX is the first line of defense in plants that neutralizes ROS and reflects how plants respond to nCS-I treatments as biostimulants [27].

### 2.3. Photosynthetic Pigments and Total Chlorophyll

The results of the evaluation of the photosynthetic pigments are shown in Figure 3, revealing differences between the treatments. In the evaluation of the chlorophyll a content (Figure 3a) at the first sampling (16 DAT), the application of KI-5, nCS-KI-25, and nCS-KIO_3_-5 exceeded the absolute control by 44.69, 44.09, and 41.51%, respectively. At 46, 76, and 106 DAT, no significant differences were observed compared with the absolute control. Compared with the absolute control, the application of nCS-KI-25, KI-5, and nCS-KIO_3_-5 increased the chlorophyll b content (Figure 3b) by 36.42, 35.05, and 32.64%, respectively, at 16 DAT. At 46, 76, and 106 DAT, no significant differences were observed with respect to T0. For the carotenoid (Figure 3c) content at only 76 DAT, significant differences were observed with respect to the absolute control, with the contents of nCS-KIO_3_-25 and nCS-KIO_3_-5 increasing by 58.88 and 54.44%, respectively.

Figure 4 shows the total chlorophyll content, and compared with that of the absolute control, the contents of KI-5, nCS-KI-25, and nCS-KIO_3_-5 increased by 41.74%, 41.74%, and 38.80%, respectively, at 16 DAT. In the remaining samples (46, 76, and 106 DAT), no significant differences were observed.

### 2.4. Gene Expression

For the gene expression analysis, only the samples from 16 DAT were considered. The results are shown in Figure 5. Compared with the absolute control, the expression of the APX gene in KIO_3_-5, nCS, nCS-KIO_3_-25, and KIO_3_-25 increased APX expression by 0.94, 0.56, 0.48, and 0.20-fold, respectively. SOD gene expression increased compared with that of the absolute control after the application of nCS-KI-5, KI-5, nCS-KIO_3_-25, nCS-KIO_3_-5, and KIO_3_-25 by 6.92, 1.75, 0.66, 0.39, and 0.11-fold, respectively. The results of the GPX gene analysis revealed that compared with the absolute control, the KI-5, KI-25, nCS-KI-5, and KIO_3_-25 treatments increased GPX expression by 38.44, 20.98, 1.02, and 0.19-fold, respectively. In addition, nCS-KIO_3_-25 and nCS-KI-25 increased by 0.14-fold with respect to the absolute control. The expression of the CAT gene with respect to the absolute control increased with the application of nCS-KIO_3_-5 by 1.39-fold, with nCS-KIO_3_-5, it increased by 1.34-fold, and with KIO_3_-25, it increased by 1.13-fold.

### 2.5. Mineral Content

The results of the macromineral content quantified in the stems, leaves, and fruits of tomato plants treated with nCS-I are shown in Table 1. No significant differences were observed with respect to the absolute control, which could be associated with an effect proportional to the concentration of iodine present in the complex. Table 2 shows the micromineral content results. The results revealed that there were no adverse effects on fruits, stems, or leaves, with no significant differences found.

### 2.6. Iodine Content

Figure 6 presents the results of the iodine content of the stems, leaves, and fruits of the tomato plants treated with nCS-I. The content of this element did not increase significantly in the three parts of the plants evaluated.

### 2.7. Shelf Life Under Cold Conditions

Table 3 shows the results of the quality parameters of tomato fruits. Compared with the control, the treatments did not significantly affect the pH, electrical conductivity (EC), or firmness of the fruits. However, in terms of the oxidation-reduction potential (ORP), we observed significant differences with respect to the absolute control in fruits treated with nCS-KIO_3_-5 and nCS-KIO_3_-25, which presented the best results, ranging from 160.00 to 161.67 mV. Compared with those in the absolute control, the total soluble solids in the KIO_3_-5 and KIO_3_-25 treatments decreased to 41.88% and 46.20%, respectively.

Figure 7 shows the percentage of weight loss over the 20 days of evaluation. In terms of weight loss, the same trend was observed as that in the development of the fruits during cold post-harvest conditions. The results did not indicate significant differences between treatments.

Figure 8 shows the results of the biochemical parameters associated with the nutritional quality of the tomatoes, including the variables of vitamin C and lycopene, and no treatment increased with respect to the absolute control. In terms of the total phenol content (Figure 8b), significant differences were found between the absolute control and the application of nCS-KIO_3_ 25 mg L^−^^1^, which increased the total phenol content by 5.40-fold, and that of nCS-KI-5, which increased the total phenol content by 4.81-fold. The application of KI-25 and nCS increased the antioxidant capacity (Figure 8d) by 12.85% and 10%, respectively, compared with that of the absolute control.

## 3. Discussion

### 3.1. Total Protein

Plants under oxidative stress undergo remarkable changes at the proteomic level, including total protein content, as well as post-transcriptional and post-transcriptional modifications, protein–protein interactions, and ultimately biological functions of proteins aimed at metabolic adjustments to a changing environment and an improvement of plant tolerance to stress [28]. The results obtained are within the expected range in tomato plant leaves [29,30].

The phenomenon observed in the application of nCS-I as well as the application of the different iodine salts can be associated with greater assimilation of the treatments, which leads to the production of secondary metabolites as well as the promotion of proteins necessary for metabolism [31]. Regarding the above, a correlation analysis of the total protein content with the variables analyzed was performed, and negative correlations were found, so the results are considered consistent with the fact that low-stress levels allow for greater protein accumulation while higher stress levels result in an increase in the antioxidant activity of the enzymes SOD, CAD, APX, and GPX [32]. Following Davila Rangel et al. [25], we consider values greater than or equal to 0.8 to be useful; however, when running the correlations, significant values with negative signs less than 0.5 were found. This indicates that the protein concentration is negatively correlated with the antioxidant activity of the enzymes.

Some authors have observed positive effects on nCS assimilation with exogenous applications of KI and KIO_3_, which could be related to an increase in protein synthesis, which is an essential component of amino acids, the basic blocks that form proteins [22]. In agreement with our study, Blasco et al. [33] observed in lettuce plants significant increases in total protein content with the application of 80 µM iodine as IO_3_. In addition, Behboudi et al. [34] showed an increase in the photosynthetic rate of the corn crop reflected in the increase in grain protein with the application of chitosan nCS at 90 ppm. Regrettably, a limited amount of research has been conducted on the impact of nCS-I on plant metabolic processes [35].

### 3.2. Antioxidant Enzymatic Activity

Significant differences in enzymatic activity were observed after the application of the treatments, demonstrating that the doses of iodine applied induced a response in the antioxidant metabolism of tomato plants. GPX is an essential antioxidant enzyme in plants that plays an important role in protection against oxidative stress, and the results of GPX activity are in agreement with those of Hassan et al. [36], who reported that N can induce the enzymatic antioxidant system and (hydrogen peroxide) H_2_O_2_ scavenging in *Catharanthus roseus* (L.) G. due to the increase in the levels of antioxidant enzymes such as CAT and APX. The application of nCS has been shown to have an effect on antioxidant metabolism; for example, Sen et al. [37] reported that, in mung beans subjected to salinity stress, a positive effect on antioxidant enzymes was detected, and the priming of nCS application was demonstrated by reduced content of malondialdehyde (MDA) and H_2_O_2_.

The APX enzyme plays a crucial role in plants as part of their antioxidant system, and increased APX activity promotes plant defense mechanisms [38]. We observed an increase in the activity of this enzyme with the application of KI, and the same phenomenon of increasing APX was observed by Halka et al. [39] with the application of KI at doses of 25 and 50 μM of iodine in young tomato plants. Medrano-Macías et al. [22] reported that inorganic iodine forms (I^−^ and IO_3_) had no effect on APX activity in tomato seedlings. In contrast, research by Blasco et al. [40] on lettuce plants revealed that applying IO_3_ at doses of 20, 40, and 80 μM increased APX activity in the leaves.

The results of the present investigation show that with the application of nCS, it was possible to increase the activity of the enzymes. Similar results were reported by Thuy et al. [41] for increasing CAT activity with 0.5% nanochitosan and enzyme content of 0.049 U min^−^^1^g^−^^1^ in the rice cultivar OM18 under salinity stress, with the authors reporting that the demand for catalase synthesis to detoxify rice cells was very high. Caudhary et al. [42] reported an increase in catalase activity (60.09%) with the application of nanochitosan and *Bacillus* spp. in *Zea mays* under field conditions.

Chandra et al. [43] demonstrated that with applications of nCS on the leaves of Camellia sinensis, the enzymatic activity of SOD and CAT was increased by 41 and 49%, respectively, 24 h after the application of the treatments compared with the control. In addition, Blasco et al. [33] demonstrated that the application of 80 µM KI led to a decrease in SOD activity in lettuce grown hydroponically. The results of this study demonstrated that the applied treatments effectively increased the activities of catalase (CAT), glutathione peroxidase (GPX), and superoxide dismutase (SOD). These enzymes are functionally interconnected, as the reaction product of SOD, H_2_O_2_, serves as a substrate for CAT and GPX [44]. In this sense, depending on the source of I, the potential catalytic activity of antioxidant enzymes can increase in response to stress tolerance in plants [22]. Some studies have shown that applying KIO_3_ at concentrations of 20, 40, and 80 µM directly to the substrate increases the enzyme activities of SOD and APX [45]. This means that iodine can affect the activities of various proteins via protein iodination, which improves the modulation of the activity of antioxidant enzymes, helping plants counteract the oxidative stress generated by ROS [35]. The specific physiological mechanism underlying this phenomenon remains unclear; however, it may be attributed to the broad redox capacity of the element [46]. According to Medrano-Macías et al. [31], the effects of iodine application are based on two primary mechanisms: (1) a direct reaction between reduced iodine and reactive oxygen species (ROS), functioning as an inorganic antioxidant, which is most evident in aquatic species such as brown algae, and (2) its role as a pro-oxidant, stimulating increased antioxidant synthesis.

### 3.3. Photosynthetic Pigments and Total Chlorophyll

Photosynthesis is a key metabolic process and can be directly associated with the growth and productivity of green plants [42]. Chlorophyll molecules play a pivotal role in converting absorbed solar radiation into stored chemical energy while mediating the exchange of matter and energy flows between the biosphere and the atmosphere [47]. In this study, the treatments evaluated had a change in chlorophyll content compared with the controls, demonstrating that the application of nCS-I complexes and iodine in the form of KI and KIO_3_ can improve the photosynthetic activity of tomato plants at specific times of crop development. In agreement with our results, Medrano-Macías et al. [31] reported a positive correlation with the increase in chlorophyll a and b attributed to the antioxidant of iodine application (KIO_3_) in strawberry plants under normal and salinity stress. On the other hand, negative effects have been reported with applications of iodine that generate intracellular oxidation or the loss of electrons that convert it into molecular iodine, which can attach to cellular components, such as chlorophyll [48]. Although information on the mechanism of toxicity or transport of iodine within plants is limited, plants can increase the content of organic compounds (such as carotenoids, ascorbic acid, and polyphenols) in relation to the low concentration of available iodine in the emerged areas [46].

In the application of nCS-I, no effect on photosynthetic pigments has been observed [25]; however, in the study of the effect of nCS, it has been found that it can positively impact the photosynthetic rate of plants, as observed in this study. In agreement with our results, Choudhary et al. [49] reported an increase in chlorophyll content following the application of 1% nCS to maize. In contrast, Balusamy et al. [50] mentioned that under salt stress conditions, the exogenous application of nCS in milk thistle plants can decrease photosynthetic pigments chlorophyll a and b and total chlorophyll.

### 3.4. Gene Expression

As sessile organisms, plants are constantly exposed to various environmental stimuli that lead to rapid changes in the production and elimination of reactive oxygen species, such as H_2_O_2_ [51]. In this sense, the induction of the plant defense mechanism to avoid ROS is related to the expression of genes that encode key proteins for the immune system and the synthesis of both enzymatic and nonenzymatic compounds [52]. In the present experiment, there was differential expression of all of the genes studied with the application of nCS-I and iodine salts with respect to the absolute control.

In the *APX* gene, the results are the product of the application of nCS-I complexes and KIO_3_ salts at different concentrations that can trigger a series of signals that lead to the expression of defense genes, which, in turn, encode proteins that directly or indirectly mitigate the adverse effects of oxidative stress [53]. Some authors, such as Javed et al. [54], associated a similar phenomenon in the relative expression of *APX* gene expression and ROS scavenging enzyme (APX) activity with a greater response to stress in the cultivation of *Moringa oleifera*. In this sense, Pour-Aboughadareh et al. [55] noted that under conditions of water stress, the overexpression of this gene is related to the balance between the generation and detoxification of ROS at the intracellular level. This coincides with the findings of Qu et al. [53], as increases in APX activity essentially help eliminate intracellular levels of H_2_O_2_. Furthermore, *APX* gene expression may be part of a signaling network that activates additional defense responses, coordinating plant resistance to stress, improving growth, and increasing the accumulation of antioxidant compounds, which positively impacts the nutraceutical quality of the fruit [51].

*SOD*s are a family of genes whose enzymatic products can actively dismutase superoxide radicals (O_2_^−^) in various plant organelles [56]. In this study, the evaluated treatments led to increases in *SOD* gene expression. Shams et al. [57] mentioned that this phenomenon is associated with an increase in the expression of genes that allow it to fulfill its primary function to protect cells from oxidative stress by catalyzing the dismutation of O_2_^−^ to H_2_O_2_. Some studies have shown that the application of nCS can alleviate salt stress damage by regulating genes that mediate plant innate immune modulation [49]. In this sense, Chun and Chandrasekaran [58] with applications of nCS in tomato crops under *Fusarium andiyazi* stress, reported that the overexpression of the *SOD* gene was associated with better tolerance to stress. Hernández-Hernández et al. [59], in applications of nCS-Cu in tomato plants under salt stress, reported a 0.9-fold increase in the expression of the *SOD* gene, which was associated with activation to eliminate ROS. The application of iodine has been shown to improve stress tolerance by stimulating antioxidant mechanisms; however, little has been studied in genetic terms [22].

There were significant differences observed in *GPX* gene expression after the application of nCS-I complexes and iodine salts; this scenario may be due to cell signaling mechanisms in plants, which are triggered by signaling molecules produced within the plant [5]. However, the increase in the expression of *GPX* genes represents a biomarker of intracellular oxidative stress, which could be associated with the treatments applied on the basis of iodine in the form of KI [60]. Furthermore, the overexpression of this gene may be related to its ability to reduce H_2_O_2_ to H_2_O (using ascorbate as an electron acceptor) and promote plant growth [61]. The overexpression of *GPX* decreases oxidative damage to cell membranes, which improves the ability of plants to maintain normal metabolic function [62]. The tolerance of Arabidopsis to metal or metalloid (Al, Zn, Cd, As, and Cu) toxicity stress is increased when *APX* and *SOD* genes are upregulated [63].

On the other hand, in *CAT* gene expression on the basis of the observed behavior, the treatments applied in the presence of KIO_3_ and nCS-KIO_3_ have an indirect effect on the resistance of the plants, given that the treatments could induce the biosynthesis of various enzymatic or nonenzymatic compounds involved in the plant’s response to environmental stress [23]. Some authors have shown that the increase in *CAT* gene expression occurs in response to numerous abiotic stress factors through the elimination of H_2_O_2_ which is subsidized for ROS homeostasis [64]. In this sense, Yang et al. [65] reported that plant immune mechanisms increase *CAT* gene expression to reduce ROS accumulation in tomato plants through abiotic stress. Likewise, Zhang et al. [66] also reported on *CAT* gene expression that was altered by the interaction between catalase and biotic stress factors. ROS are generated as products of cellular metabolism, especially during situations of abiotic and biotic stress, and *CAT* helps keep H₂O₂ levels under control, preventing their toxic accumulation and minimizing oxidative damage to DNA, proteins, and lipids [67]. According to our transcriptomic data, the increase in the expression of antioxidant genes plays a key role in defense and attraction mechanisms in response to the environment [35,44].

### 3.5. Mineral Content

In this study, mineral content was quantified in the leaves, stems, and fruits; however, we did not observe a significant effect with the application of the treatments, which shows that no negative effects on the mineral content were induced. The biostimulation of crops through agronomic practices via the exogenous application of nCS is a practice that favors the nutritional quality of and yields of crops [42]. Similarly, the exogenous application of iodine sources in plants can improve osmotic balance and nutrient transport [24]. There were no significant differences in the quantification of macro and microelements among the treatments (Table 1 and Table 2). Dávila Rangel et al. [25], with direct applications to the substrates of CS-KI and CS-KIO_3_ (5 and 25 mg iodide kg^−1^, respectively) in lettuce cultivation, reported a trend similar to that reported in this study, relating this behavior to the change in redox equilibrium caused by iodine during cultivation. In lettuce, soil application of KI (0.5–2.0 kg ha^−1^) and foliar spraying with KIO_3_ (0.02–2 kg ha^−1^) did not significantly alter the mineral composition of lettuce plants [22].

### 3.6. Iodine Content

In the evaluation of iodine content, no significant differences were observed between treatments; however, a different range of content was observed depending on the organ of study (leaves, stems, and fruits). In this sense, the recommended daily intake of iodine is 90 to 200 µg depending on the age and stage of development. According to the World Health Organization (WHO) and the United Nations Children’s Fund (UNICEF), this dose can be covered with the intake of one or two tomato fruits produced under the evaluated conditions and according to the evaluated treatments.

On the other hand, this element is not considered a macro- or micronutrient; however, it plays a key role in the production of antioxidants, adaptation to new environments, and improved performance [31]. The absorption, transport, and accumulation of iodine are influenced by the concentration and ionic form of the element in the environment. However, in some species, the majority of this nutrient is retained in the roots [68]. Medrano-Macías et al. [22] mentioned that in the exogenous application of iodine, no significant differences could be observed between the different forms of iodine applied (KI^−^ and KIO_3_). This phenomenon can be attributed to the nature of the element since its volatilization is increased by external environmental factors (temperature, humidity, etc.) that accelerate biochemical reactions and limit its translocation in plants [69]. This could partly explain the low levels observed in the plants because, during the development of the crop, there were external conditions (temp.: 35 to 45 °C) that could generate a source of stress to the plants as well as increase the volatilization of iodine.

On the other hand, Cortés-Flores et al. [70] reported a positive correlation between the iodine content applied and that absorbed by bell pepper seedlings. However, the treatments applied in this research followed the classification of a biostimulant, so the range of application compared with that in other studies was low. For example, as described by Kiferle et al. [71], in irrigation applications of KI at concentrations of 1, 2, and 5 mM, KIO_3_ at concentrations of 0.5, 1, and 2 mM significantly accumulated in tomato fruits, achieving biofortification of the crop while maintaining sufficient iodine levels when the crop was consumed. According to Medrano-Macías et al. [22], interactions between mineral elements can be either synergistic or antagonistic. Synergistic interactions lead to increased absorption, transport, uptake, or metabolism of an element in the presence of iodine, whereas antagonistic interactions result in a reduction in these activities when iodine is present. However, in this study, even though the mineral content did not increase with the application of treatment, a positive effect was observed in improving the antioxidant metabolism of tomato plants.

### 3.7. Shelf Life Under Cold Conditions

Currently, there is no information on the impact of the application of nCS-I complexes on the nutritional and organoleptic characteristics of tomatoes postharvest through biostimulation. However, the application of iodine to crops at adequate concentrations has a biostimulant effect since it promotes the synthesis of enzymatic and nonenzymatic antioxidants in plants [71]. Regarding the shelf life of fruits treated with nCS-I and iodine salts, differences were only observed in the variables of ORP, total phenols, and antioxidant capacity. In terms of the ORP results, positive values indicate that oxidizing agents, such as free radicals or compounds derived from oxidative processes, predominate, which can be explained by the climacteric nature of the fruit, as well as the degree of maturity or the deterioration process [72]. Therefore, treatment with iodine salts and nCS did not affect the weight of the fruits. This finding coincides with what was reported by Kiferle et al. [71], who reported that biofortification with KI and KIO_3_^−^ had no effect on the dry weight of tomato fruits. On the other hand, in terms of the biochemical parameters, significant differences were found among the treatments; this phenomenon can be attributed to the interaction of the plant with the complex and iodine salts, which increased the synthesis of nonenzymatic compounds, promoting the accumulation of phenolic compounds and antioxidant capacity that improve the nutritional quality of the fruits [73]. Blasco et al. [74] reported that under saline stress, iodine increases foliar mass in lettuce, the antioxidant response, and the accumulation of phenolic compounds at 20 and 40 µM KIO_3_. However, there is limited information on the scope of treatments through the biostimulation process in the shelf life of fresh fruits such as tomato.

In this study, the application of salts and the nanochitosan complex as a control increased the antioxidant capacity of the fruits, indicating that tomatoes are more resistant to the negative effects of ROS, such as lipid peroxidation and damage to proteins and DNA [38]. This helps maintain the cellular integrity of the fruit.

## 4. Materials and Methods

### 4.1. Experimental Conditions and Plant Material

The experiment was performed in a polyethylene-covered greenhouse at the Horticulture Department of the Universidad Autónoma Agraria Antonio Narro (Saltillo, Mexico) from May–October 2022. CID F1 hybrid tomato seeds (Harris Moran Seed Company, Modesto, CA, USA; saladette type and indeterminate growth) were sown in 200-well polystyrene trays. When the seedlings reached an approximate size of 15 cm, they were transplanted into polyethylene bags with a capacity of 10 kg, and a 1:1 ratio of peat moss to perlite was used as a substrate. The nutrient mixture was supplied according to the plant’s water requirements using the method of Steiner [75], starting at 25% after transplanting until the plants had nine pairs of leaves with 5 irrigations during the day of 50 mL each. After that, it was then increased to 50% until the beginning of flowering with 10 irrigations during the day of 150 mL each, reaching 100% at the end of the experiment with 25 irrigations during the day of 250 mL each. In addition, at night, supplementary irrigation of 2000 mL per pot was performed. To prevent the appearance of pathogens, three foliar applications of Captan 50 WP were made during the transplant, development, and flowering stages. Chromatic traps were also used to control insects. The average temperature and relative humidity (RH%) of the greenhouse were evaluated at 15-day interval readings, giving 8 readings during the 120 days of the crop cycle (Table 4). The values were recorded with a digital thermohygrometer and 98 μmol m^−2^ s^−1^ PAR (photosynthetically active radiation). The duration of the experiment was 120 days after transplant (DAT).

### 4.2. Treatments

The experiment evaluated 10 treatments with five experimental units composed of four plants per unit (Figure 9). The treatments used are described below: T0: absolute control; nCS: nanochitosan as a control; KIO_3_-5 mg L^−^^1^; KIO_3_-25 mg L^−^^1^; nCS-KIO_3_ complex with 5 mg L^−^^1^ of iodate; nCS-KIO_3_ complex with 25 mg L^−^^1^ of iodate; KI-5 mg L^−^^1^; KI-25 mg L^−^^1^; nCS-KI complex with 5 mg L^−^^1^ of iodide; and nCS-KI complex with 25 mg L^−^^1^ of iodide. The synthesis of iodinated nanochitosan complexes is detailed in the protocols established by Ortega-Ortiz et al. [76] and Rivera-Solís et al. [77]. Eight applications were made every 15 days from the time of transplantation, which lasted 120 days. Foliar application was performed at the beginning of the day under conditions of relative humidity and air temperature suitable for promoting foliar absorption. To avoid contamination from neighboring treatments, a flexible plastic barrier was used to isolate the application space.

### 4.3. Sampling and Evaluation of Variables

Four leaf samples were collected 24 h after the application of the treatments at specific times: (1) second application of the treatments (16 DAT), fourth application of the treatments (46 DAT), sixth application of the treatments (76 DAT), and eighth application of the treatments (106 DAT). For each sampling stage, three leaf samples composed of two randomly selected plants were collected. These samples were taken by collecting the third apical leaf of the plant, immediately freezing it with liquid nitrogen, and storing it in an ultrafreezer at −80 °C to determine stress-indicating metabolites and the expression of defense genes, and some of the samples were lyophilized for some determinations, such as enzyme activity. From each treatment, all of the fruit per plant per replicate were harvested, from the first to the sixth bunch of tomatoes, when the fruit showed an intense red color, according to the USDA [78]. In addition, a shelf life test was established with fruits harvested from the third cluster using five fruits per sampling time under cold temperature (4 °C) conditions for 20 days. At the end of the crop cycle, the mineral content was analyzed via composite samples taken from leaves, stems, and fruits. For this purpose, two plants were randomly selected per treatment, with three replicates for each.

### 4.4. Biochemical Analysis

To detect stress-related metabolites in tomato plant leaves, previously frozen tissues were freeze-dried, manually ground using a mortar and pestle, and subsequently analyzed biochemically. A UV-Vis spectrophotometer (Thermo Scientific Model G10S, Waltham, MA, USA) was used for biochemical analysis. The Bradford method [79] was used to measure the total protein content. SOD activity was measured following the methodology described by Medrano-Macias et al. [2], using the biomolecule extract and the Cayman^®^ 7,060,002 commercial assay kit (Cayman Chemical Company, Ann Arbor, MI, USA); the results are expressed in U mL^−^^1^. Catalase activity was quantified via spectrophotometry, and two reaction times (T0 and T1) were recorded, following the methodology described by Dhindsa et al. [80]; the results are expressed in mM H_2_O_2_ per total protein content. GPX activity was measured as described by Flohé and Günzler [81] and was expressed as mM glutathione per minute per total protein content. The activity of APX in the biomolecule extracts was determined according to the spectrophotometric technique established by Nakano and Asada [82], considering two reading times (T0 and T1). The values of APX were expressed as enzyme activity (1 μmol of substrate (ascorbic acid) converted per minute per total protein content). Photosynthetic pigments were determined via the method described by Wellburn [83], with some modifications: Chl a: chlorophyll a; Chl b: chlorophyll b; total C: total chlorophyll; and Car: carotenoids. For all of the enzymatic and photosynthetic pigment analyses, five composite replicates were considered, with two leaves per replicate per treatment.

### 4.5. Real-Time Reverse Transcription PCR

For the analysis of gene expression, three replicates composed of two leaves each were considered per treatment. RNA was extracted via TRI Reagent^®^, purified with chloroform, and precipitated with isopropanol, as described by Cui et al. [84]. The quantification of DNase I-treated RNA (Sigma-Aldrich, Burlington, MA, USA) was performed via a UV-Vis spectrophotometer with an A260/A280 nm ratio, and the quality was determined visually via denaturing electrophoresis. cDNA synthesis was performed via a commercial Bioline kit (SensiFAST cDNA Synthesis Kit, Toronto, ON, Canada). Actin as endogenous (*ACT*) and four genes were used: *CAT*, *SOD*, *GPX*, and *APX*. The primers used were designed via Primer BLAST tool (National Center for Biotechnology Information (NCBI), Bethesda, Rockville, MD, USA) and Oligoanalyzer 3.1 (Integrated DNA Technologies IDT, Coralville, IA, USA). The sequences of the primers used are described in Table 5.

Real-time PCR was performed on an Applied Biosystems StepOne™ version 2.3 (Thermo Fisher Scientific, Waltham, MA, USA) via the ∆∆Ct method by measuring the fluorescence intensity of SYBR™ Select Master Mix (Applied Biosystems, Foster City, CA, USA). PCR was performed at a volume of 20 µL for all genes (10 µL of Master Mix, 1 µL of cDNA, a defined primer concentration, and nuclease-free water). For the *ACT* gene, the forward primer concentration was 72 nM, and 60 nM was used for the reverse primer. For the *APX* and *SOD* genes, the primer concentrations were 300 nM. For the *GPX* and *CAT* genes, the forward primer concentration was 100 nM. Real-time PCR was performed under the following conditions: 10 min at 95 °C and 40 cycles of 15 seg at 95 °C and 1 min at 60 °C. In addition, a melt curve was generated.

### 4.6. Mineral and Iodine Contents

The mineral content was determined in the leaves, stems, and fruits of composite samples from two plants, with three replicates per treatment. The contents of K, Mg, Ca, Na, Mn, Zn, and Fe were evaluated via atomic absorption spectrometry after wet digestion on a Varian AA-1275 flame atomic absorption spectroscope (Palo Alto, CA, USA) [85]. For the determination of iodine content, the alkaline ash technique was followed [86,87]. Quantification was performed via an Agilent 725 ICP-OES (inductively coupled plasma–optical emission spectrometry, Santa Clara, CA, USA).

### 4.7. Shelf Life Analysis

Shelf life analyses were performed on tomato fruits under cold storage conditions (6 ± 2 °C and 90% relative humidity (RH)). Fruits harvested from the third bunch were evaluated at the red maturity point 87,807 (70% of the fruits presented the typical light red color of maturity) [88]. Physicochemical analyses were performed at regular intervals (0, 5, 10, 15, and 20 days) for 20 days for cold storage using five fruits per sampling time. The variables evaluated were measured following the methods described by Yadav and Singh, [89] weight loss was calculated from the difference between the initial and final weights and expressed as % weight loss; total soluble solids (TSS) were determined via a PR-101ATAGO PALETTE digital hand-held refractometer; and fruit firmness (FF) was determined via a manual penetrometer (Wagner Instruments, model FDK 20, Greenwich, CT, USA). Three measurements were taken at different points on the fruit, and the average was obtained. The results of firmness were expressed in kg cm^−^^2^; hydrogen potential (pH) and electrical conductivity (EC) were determined with a HANNA^®^ brand digital conductivity meter in mS cm^−^^1^. In addition, biochemical parameters were evaluated at the first sampling time using a fresh sample to analyze the content of vitamin C [90], total phenols [91], and lycopene [92] and antioxidant capacity [93].

### 4.8. Statistical Analysis

A completely randomized experimental design was used, with five repetitions for enzymatic activity and photosynthetic pigments and three repetitions for gene expression, mineral content, and shelf life. Statistical analysis of the results was performed via one-way analysis of variance (ANOVA) with Infostat software (v2020) to determine the differences between the absolute control and treatments. Comparisons of means were performed via Fisher’s LSD test. All evaluations were performed at a significance level of 5% (*p*-value < 0.05). Statistically significant differences are indicated with different letters in the graphs. For gene expression, a heatmap was generated with the statistical software OriginPro 2024b.

## 5. Conclusions

In the present study, the application of iodine salts and complexes (nCS-KI or nCS-KIO_3_) under stress-free conditions had a biostimulation effect on tomato plants.

SOD, CAT, GPX, and APX activities increased at crucial times during crop development with the application of nanochitosan, iodine salts (5 and 25 mg L^−^^1^), and the nanochitosan–iodine complex with 5 and 25 mg L^−^^1^ of iodate.

In addition, the expression of the antioxidant genes *SOD*, *CAT*, *GPX*, and *APX* increased with the application of nanochitosan, potassium iodate at 5 and 25 mg L^−^^1^, and the nanochitosan–iodine complex with 25 mg L^−^^1^ of iodate. These findings indicate that these treatments promoted the plant’s antioxidant system, which could be an alternative to mitigate the negative effects of oxidative stress.

The evaluated treatments did not affect the mineral or iodine content, demonstrating that the application of the treatments did not affect mineral absorption.

The application of potassium iodide at 5 mg L^−^^1^, the nanochitosan–iodine complex with 5 mg L^−^^1^ of iodide, and the nanochitosan–iodine complex with 5 and 25 mg L^−^^1^ of iodate increased the chlorophyll a and b and total chlorophyll contents of the photosynthetic pigments.

The nanochitosan–iodine complex with 25 mg L^−^^1^ of iodate and the nanochitosan–iodine complex with 25 mg L^−^^1^ of iodide promoted an increase in phenolic compound content and antioxidant activity in tomato fruits during shelf storage, which considerably impacts the nutritional quality of the fruit.

There are no similar reports on crop biostimulation with foliar-applied nanochitosan–iodine complexes. Further biostimulation studies with nanochitosan–iodine complexes and iodine salts in crop species other than the study model are needed to evaluate the effects they may have at higher doses.

## Figures and Tables

**Figure 1 plants-14-00801-f001:**
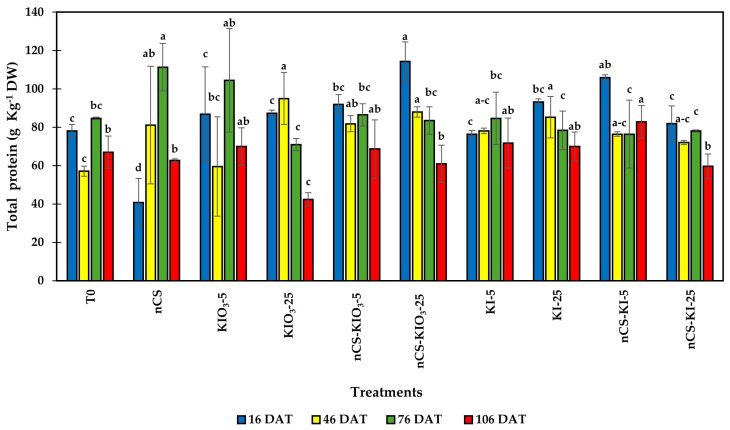
Total protein content in the leaves of tomato plants treated with nCS-I complexes and iodine salts. Different letters within each column indicate significant differences between treatments (LSD, *p* ≤ 0.05). DAT: days after transplant. DW: dry weight; n = 3; bars represent the standard deviation. T0: absolute control; nCS: nanochitosan as a control; KIO_3_-5 mg L^−^^1^; KIO_3_-25 mg L^−^^1^; nCS-KIO_3_ complex with 5 mg L^−^^1^ of iodate; nCS-KIO_3_ complex with 25 mg L^−^^1^ of iodate; KI-5 mg L^−^^1^; KI-25 mg L^−^^1^; nCS-KI complex with 5 mg L^−^^1^ of iodide; nCS-KI complex with 25 mg L^−^^1^ of iodide.

**Figure 2 plants-14-00801-f002:**
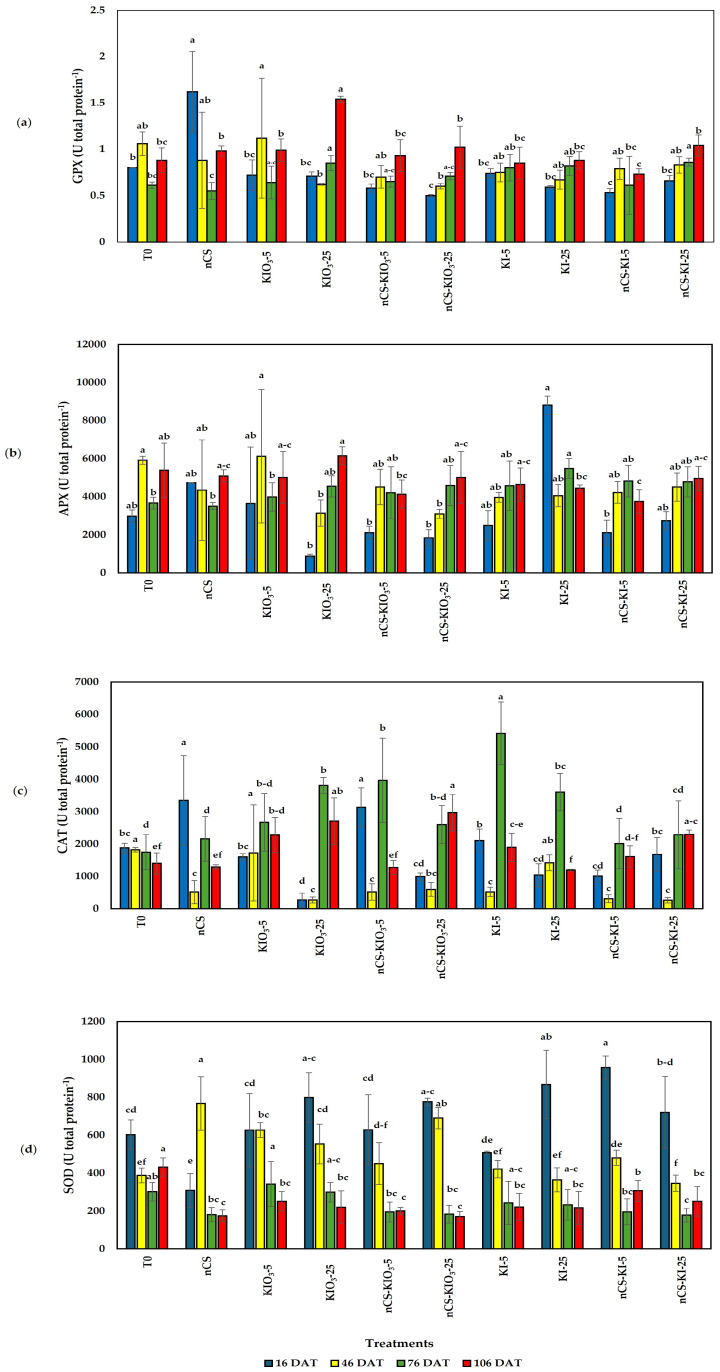
Enzymatic activity of (**a**) GPX, (**b**) APX, (**c**) CAT, and (**d**) SOD in tomato plants treated with nCS-I complexes and iodine salts. Different letters within each column indicate significant differences between treatments (LSD, *p* ≤ 0.05). DAT: days after transplant. DW: dry weight; n = 5; bars represent the standard deviation. T0: absolute control; nCS: nanochitosan as a control; KIO_3_-5 mg L^−^^1^; KIO_3_-25 mg L^−^^1^; nCS-KIO_3_ complex with 5 mg L^−^^1^ of iodate; nCS-KIO_3_ complex with 25 mg L^−^^1^ of iodate; KI-5 mg L^−^^1^; KI-25 mg L^−^^1^; nCS-KI complex with 5 mg L^−^^1^ of iodide; nCS-KI complex with 25 mg L^−^^1^ of iodide.

**Figure 3 plants-14-00801-f003:**
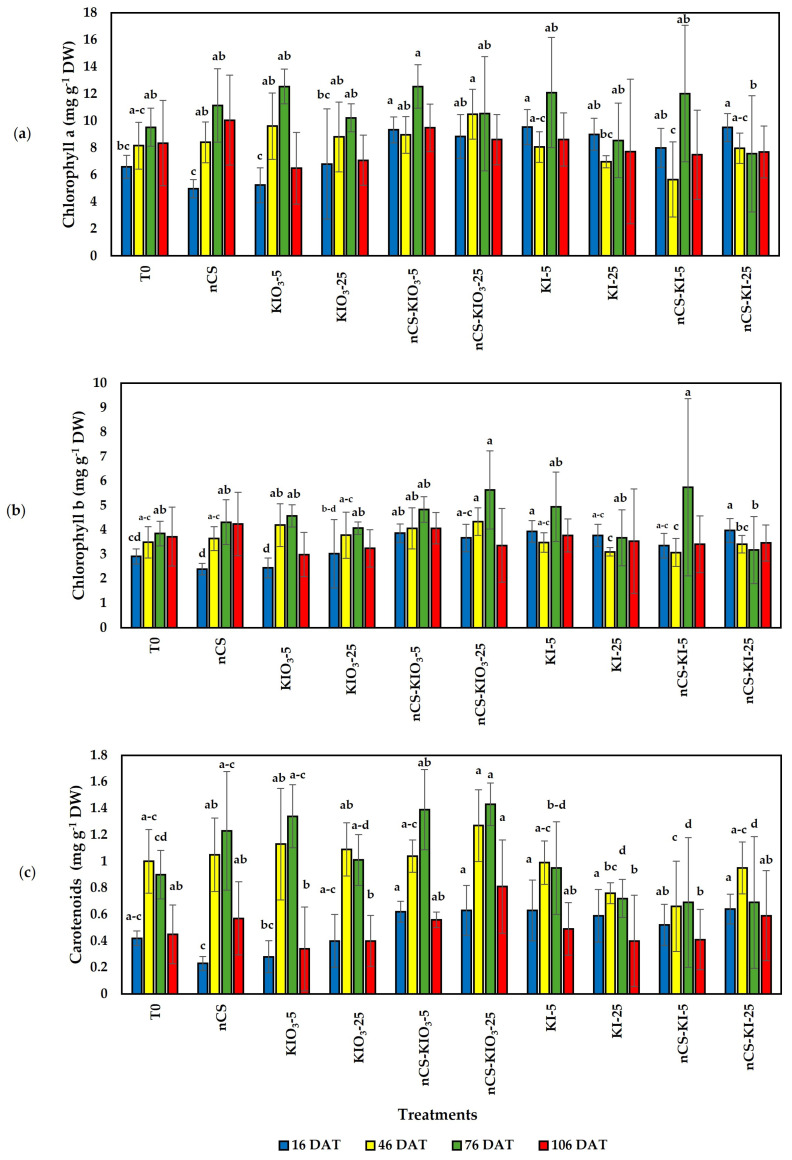
Photosynthetic pigments: (**a**) chlorophyll a, (**b**) chlorophyll b, and (**c**) carotenoids in tomato plant leaves treated with nCS-I complexes and iodine salts. No letters within each column indicate no significant differences between treatments (LSD, *p* ≤ 0.05). DAT: days after transplant; DW: dry weight, n = 5; bar intervals represent the standard deviation. T0: absolute control; nCS: nanochitosan as a control; KIO_3_-5 mg L^−^^1^; KIO_3_-25 mg L^−^^1^; nCS-KIO_3_ complex with 5 mg L^−^^1^ of iodate; nCS-KIO_3_ complex with 25 mg L^−^^1^ of iodate; KI-5 mg L^−^^1^; KI-25 mg L^−^^1^; nCS-KI complex with 5 mg L^−^^1^ of iodide; nCS-KI complex with 25 mg L^−^^1^ of iodide.

**Figure 4 plants-14-00801-f004:**
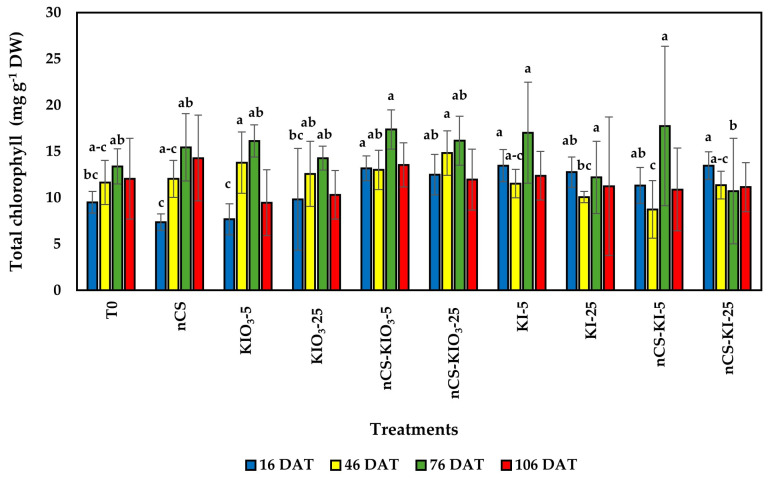
Total chlorophyll content in the leaves of tomato plants treated with nCS-I complexes and iodine salts. No letters within each column indicate no significant differences between treatments (LSD, *p* ≤ 0.05). DAT: Days after transplant; bars represent the standard deviation; n = 5. T0: absolute control; nCS: nanochitosan as a control; KIO_3_-5 mg L^−^^1^; KIO_3_-25 mg L^−^^1^; nCS-KIO_3_ complex with 5 mg L^−^^1^ of iodate; nCS-KIO_3_ complex with 25 mg L^−^^1^ of iodate; KI-5 mg L^−^^1^; KI-25 mg L^−^^1^; nCS-KI complex with 5 mg L^−^^1^ of iodide; nCS-KI complex with 25 mg L^−^^1^ of iodide.

**Figure 5 plants-14-00801-f005:**
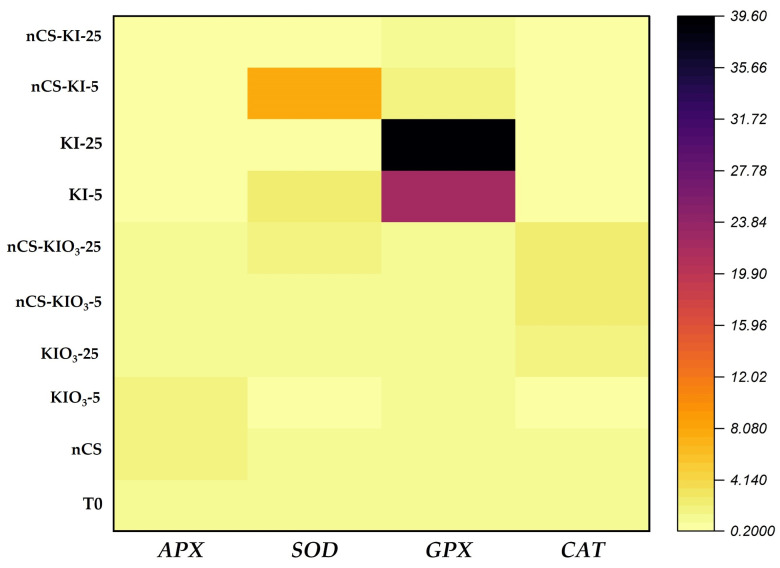
Heatmap of the relative expression of defense genes in tomato plant leaves treated with nCS-I complexes and iodine salts at 16 DAT. *APX*: ascorbate peroxidase gene; *SOD*: superoxide dismutase gene; *GPX*: glutathione peroxidase gene; *CAT*: catalase gene. n = 3. T0: absolute control; nCS: nanochitosan as a control; KIO_3_-5 mg L^−^^1^; KIO_3_-25 mg L^−^^1^; nCS-KIO_3_ complex with 5 mg L^−^^1^ of iodate; nCS-KIO_3_ complex with 25 mg L^−^^1^ of iodate; KI-5 mg L^−^^1^; KI-25 mg L^−^^1^; nCS-KI complex with 5 mg L^−^^1^ of iodide; nCS-KI complex with 25 mg L^−^^1^ of iodide.

**Figure 6 plants-14-00801-f006:**
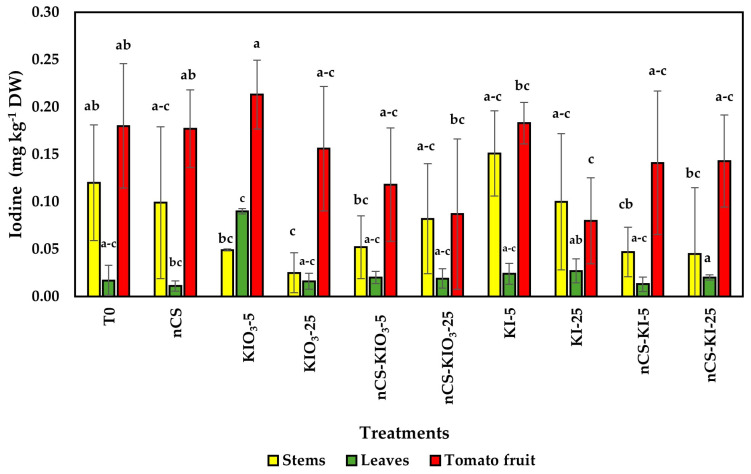
Iodine concentrations in tomato plants treated with nCS-I complexes and iodine salts. Means with the same letter are statistically identical (LSD, *p* ≤ 0.05); bars represent the standard error (n = 3). T0: absolute control; nCS: nanochitosan as a control; KIO_3_-5 mg L^−^^1^; KIO_3_-25 mg L^−^^1^; nCS-KIO_3_ complex with 5 mg L^−^^1^ of iodate; nCS-KIO_3_ complex with 25 mg L^−^^1^ of iodate; KI-5 mg L^−^^1^; KI-25 mg L^−^^1^; nCS-KI complex with 5 mg L^−^^1^ of iodide; nCS-KI complex with 25 mg L^−^^1^ of iodide.

**Figure 7 plants-14-00801-f007:**
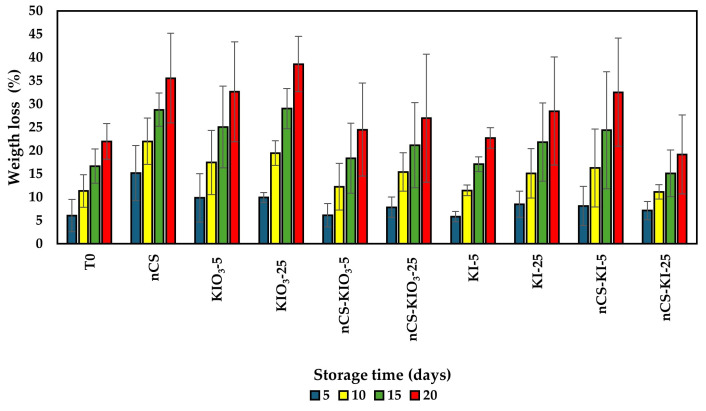
Post-harvest weight loss percent of tomatoes treated with nCS-I complexes and iodine salts under cold conditions. Means with no letter are statistically identical (LSD, *p* ≤ 0.05); the bars represent the standard deviation, sample size (n = 3). T0: absolute control; nCS: nanochitosan as a control; KIO_3_-5 mg L^−^^1^; KIO_3_-25 mg L^−^^1^; nCS-KIO_3_ complex with 5 mg L^−^^1^ of iodate; nCS-KIO_3_ complex with 25 mg L^−^^1^ of iodate; KI-5 mg L^−^^1^; KI-25 mg L^−^^1^; nCS-KI complex with 5 mg L^−^^1^ of iodide; nCS-KI complex with 25 mg L^−^^1^ of iodide.

**Figure 8 plants-14-00801-f008:**
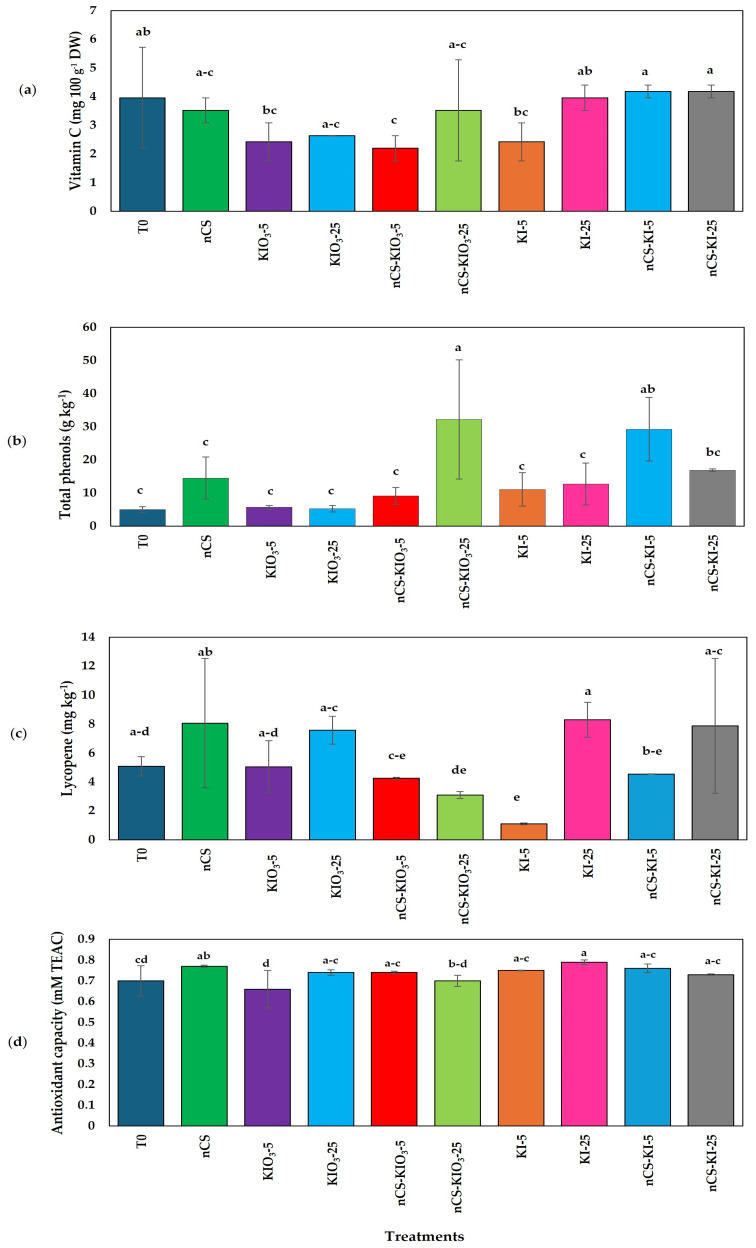
Evaluation of quality parameters: (**a**) vitamin C content, (**b**) total phenol content, (**c**) lycopene content, and (**d**) antioxidant capacity under cold postharvest conditions. Means with the same letter are statistically identical (LSD, *p* ≤ 0.05); sample size (n = 3); DW: dry weight. T0: absolute control; nCS: nanochitosan as a control; KIO_3_-5 mg L^−^^1^; KIO_3_-25 mg L^−^^1^; nCS-KIO_3_ complex with 5 mg L^−^^1^ of iodate; nCS-KIO_3_ complex with 25 mg L^−^^1^ of iodate; KI-5 mg L^−^^1^; KI-25 mg L^−^^1^; nCS-KI complex with 5 mg L^−^^1^ of iodide; nCS-KI complex with 25 mg L^−^^1^ of iodide.

**Figure 9 plants-14-00801-f009:**
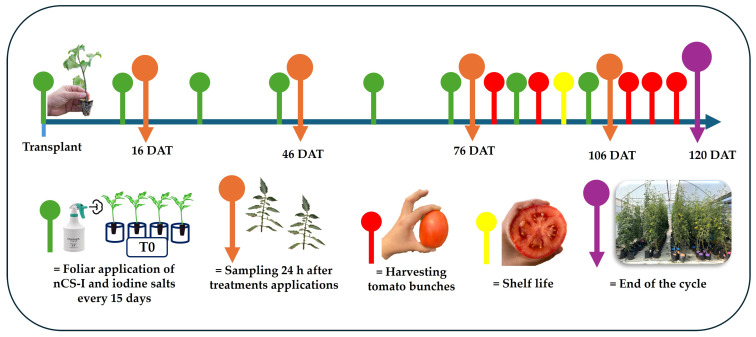
Schematic representation of the experimental development. nCS-I: iodinated nanochitosan complexes and iodine salts: potassium iodate (KIO_3_) and potassium iodide (KI); DAT: days after transplantation.

**Table 1 plants-14-00801-t001:** Macronutrient content in samples of tomato stems, leaves, and fruits treated with nCs-I complexes and iodine salts.

Organ	Treatments	mg kg^−1^, DW (n = 3)			
		K	Na	Mg	Ca
Stems	T0	7665.33 ab	101.33	1551.67	3821.67
	nCs	6752.67 ab	236.00	4780.00	3588.00
	KIO_3_-5	6325.00 ab	N.D.	1127.33	8446.33
	KIO_3_-25	4639.33 ab	N.D.	720.00	2305.00
	nCS-KIO_3_-5	4434.00 ab	N.D.	849.67	4261.67
	nCS-KIO_3_-25	8808.00 ab	332.67	6028.00	5812.33
	KI-5	4787.67 ab	N.D.	841.00	3506.00
	KI-25	4911.33 ab	N.D.	855.67	6128.33
	nCS-KI-5	2847.33 b	398.00	798.33	1729.33
	nCS-KI-25	14,650.33 a	N.D.	1851.00	7875.33
Leaves	T0	4230.67	101.00	1037.00 b	5448.33
	nCS	11,245.67	741.67	6315.67 ab	30,094.00
	KIO_3_-5	4205.00	635.67	2692.00 b	30,814.67
	KIO_3_-25	9126.67	97.00	6825.67 ab	26,208.33
	nCS-KIO_3_-5	9353.33	N.D.	6212.67 ab	21,296.00
	nCS-KIO_3_-25	1270.67	5.67	1105.00 b	10,559.33
	KI-5	13,639.67	838.67	10,472.33 a	33,691.00
	KI-25	8084.67	N.D.	3637.00 ab	18,326.00
	nCS-KI-5	11,484.33	N.D.	5537.33 ab	17,739.00
	nCS-KI-25	11,479.00	29.67	1278.00 b	24,097.00
Tomato fruit	T0	15,520.67	37.00 c	667.00	696.33
	nCS	11,626.67	547.33 ab	488.67	554.33
	KIO_3_-5	9168.33	618.00 ab	347.67	602.33
	KIO_3_-25	18,238.67	498.00 ab	564.00	763.33
	nCS-KIO_3_-5	6702.33	495.33 ab	281.33	381.67
	nCS-KIO_3_-25	14,185.33	755.67 a	557.67	519.00
	KI-5	10,490.33	510.67 ab	338.00	475.67
	KI-25	8619.67	563.67 ab	375.00	370.67
	nCS-KI-5	10,111.67	732.33 a	462.00	698.67
	nCS-KI-25	10,264.00	313.33 bc	329.33	528.67

Means with no letters indicate no significance differences (LSD, *p* ≤ 0.05); sample size (n = 3). K: potassium; Na: sodium; Mg: magnesium and Ca: calcium. N.D.: not detected. T0: absolute control; nCS: nanochitosan as a control; KIO_3_-5 mg L^−1^; KIO_3_-25 mg L^−1^; nCS-KIO_3_ complex with 5 mg L^−1^ of iodate; nCS-KIO_3_ complex with 25 mg L^−1^ of iodate; KI-5 mg L^−1^; KI-25 mg L^−1^; nCS-KI complex with 5 mg L^−1^ of iodide; nCS-KI complex with 25 mg L^−1^ of iodide.

**Table 2 plants-14-00801-t002:** Microelement contents in samples of tomato stems, leaves, and fruits treated with nCs-I complexes and iodine salts.

Organ	Treatments	mg kg^−1^, DW (n = 3)			
		Zn	Fe	Mn	Cu
Stems	T0	538.00 ab	303.67 a	70.33 a	N.D.
	nCS	446.00 ab	306.67 a	17.67 b	N.D.
	KIO_3_-5	531.33 ab	276.67 ab	20.00 b	39.67
	KIO_3_-25	488.33 ab	224.33 a–c	11.00 b	4.67
	nCS-KIO_3_-5	439.00 ab	256.67 a–c	12.00 b	7.67
	nCS-KIO_3_-25	427.33 ab	305.67 a	17.67 b	21.00
	KI-5	452.33 ab	82.33 a–c	10.67 b	N.D.
	KI-25	345.00 b	212.00 a–c	17.67 b	N.D.
	nCS-KI-5	791.67 a	61.67 bc	9.67 b	N.D.
	nCS-KI-25	402.33 ab	34.33 c	27.00 b	N.D.
Leaves	T0	259.67	1134.67	30.67	3.00
	nCS	279.00	1061.67	174.67	N.D.
	KIO_3_-5	299.33	987.33	75.33	N.D.
	KIO_3_-25	443.00	1178.00	215.00	195.33
	nCS-KIO_3_-5	497.67	925.33	190.00	N.D.
	nCS-KIO_3_-25	383.33	1209.00	39.67	N.D.
	KI-5	393.67	1378.00	235.67	N.D.
	KI-25	190.67	924.00	151.67	3.33
	nCS-KI-5	575.67	1754.33	161.33	20.00
	nCS-KI-25	221.67	991.67	202.00	0.67
Tomato fruit	T0	322.67 a	298.67 a	9.33 ab	34.33 a
	nCS	187.00 a–c	81.00 b	6.67 ab	5.00 b
	KIO_3_-5	182.33 a–c	93.67 b	5.00 ab	9.33 ab
	KIO_3_-25	305.00 ab	82.00 b	9.33 ab	9.33 ab
	nCS-KIO_3_-5	256.67 a–c	68.00 b	4.33 b	12.67 ab
	nCS-KIO_3_-25	215.00 a–c	62.67 b	7.67 ab	N.D.
	KI-5	143.00 bc	0.67 b	18.00 a	N.D.
	KI-25	127.67 c	122.67 b	6.00 ab	N.D.
	nCS-KI-5	113.33 c	20.00 b	5.67 ab	N.D.
	nCS-KI-25	114.33 c	29.00 b	4.33 b	N.D.

Means with no letters indicate no significance differences (LSD, *p* ≤ 0.05); sample size (n = 3). Zn: zinc; Fe: iron; Mn: manganese and Cu: copper. N.D.: not detected. T0: absolute control; nCS: nanochitosan as a control; KIO_3_-5 mg L^−1^; KIO_3_-25 mg L^−1^; nCS-KIO_3_ complex with 5 mg L^−1^ of iodate; nCS-KIO_3_ complex with 25 mg L^−1^ of iodate; KI-5 mg L^−1^; KI-25 mg L^−1^; nCS-KI complex with 5 mg L^−1^ of iodide; nCS-KI complex with 25 mg L^−1^ of iodide.

**Table 3 plants-14-00801-t003:** Evaluation of the post-harvest quality parameters of tomatoes treated with nCS-I complexes and iodine salts under cold conditions.

Treatments	pH	TSS	EC (µS cm^−1^)	ORP (mV)	Firmness (kg cm^2^)
T0	4.58	894.00 a	1384.67	187.33 bc	20.95
nCS	4.50	576.67 ab	880.33	184.33 a–c	4.07
KIO_3_-5	4.73	519.67 b	1226.00	179.00 ab	2.31
KIO_3_-25	4.57	481.00 b	1174.67	179.67 ab	2.66
nCS-KIO_3_-5	4.61	785.00 ab	1046.33	160.00 a	3.67
nCS-KIO_3_-25	4.66	631.67 ab	1053.67	161.67 a	2.92
KI-5	4.53	714.67 ab	1063.33	208.00 bc	4.27
KI-25	4.64	668.33 ab	1147.00	171.67 ab	3.23
nCS-KI-5	4.73	641.33 ab	1345.33	179.67 ab	4.06
nCS-KI-25	4.66	599.67 ab	1059.33	222.33 c	2.64

Means with no letter are statistically identical (LSD, *p* ≤ 0.05); sample size (n = 3). pH; TSS: total soluble solids; EC: electrical conductivity; ORP: oxidation-reduction potential. T0: absolute control; nCS: nanochitosan as a control; KIO_3_-5 mg L^−1^; KIO_3_-25 mg L^−1^; nCS-KIO_3_ complex with 5 mg L^−1^ of iodate; nCS-KIO_3_ complex with 25 mg L^−1^ of iodate; KI-5 mg L^−1^; KI-25 mg L^−1^; nCS-KI complex with 5 mg L^−1^ of iodide; nCS-KI complex with 25 mg L^−1^ of iodide.

**Table 4 plants-14-00801-t004:** Average greenhouse temperature and relative humidity.

Period (DAT)	Temperature (°C)	RH (%)
0	28.7	51.50
15	24.7	81.00
30	20.65	89.00
45	36.35	42.50
60	38.55	40.50
75	38.15	41.00
90	41.65	38.00
105	42.75	37.00
120	44.65	33.00

**Table 5 plants-14-00801-t005:** Sequences of the primers used for the *ACT*, *SOD*, *APX*, *GPX,* and *CAT* genes.

Gene	Forward Primer 5′–3′	Reverse Primer 5′–3′	Tm
*ACT*	CCCAGGCACACAGGTGTTAT	CAGGAGCAACTCGAAGCTCA	60 °C
*SOD*	TGGGAATCTATGAAGCCCAAC	AATTGTGTTGCTGCAGCTGC	60 °C
*APX*	TGACCACTTGAGGGACGTGTT	CAGAACGCTCCTTGTGGCAT	60 °C
*GPX*	ACGGAGCAAGCGACAATTGACAAC	CGATTGATTCACCGCAAAGCTCGT	60 °C
*CAT*	CCATCCAAATAATCATCAGAG	GGATAAAATAAAAATTATTTT	60 °C

Tm: melting temperature.

## Data Availability

Data are contained within the article.

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
