# Peer review of "Tomato Biostimulation with Nanochitosan–Iodine Complexes: Enhancing Antioxidant Metabolism"

_plants, 2025, doi:10.3390/plants14050801_

Round 1

Reviewer 1 Report

Comments and Suggestions for Authors

Comments are provided in the attachment. 

Reviewer 2 Report

Comments and Suggestions for Authors

I want to start by congratulating the authors for the work carried out, as it is very important research given the climate change situation we are experiencing. It is a very well structured and carried out work that analyzes all parameters in order to truly understand the influence of these applications on tomatoes. I really enjoyed reading this paper as I do similar work in another culture, so I hope it will be published soon.

The only suggestion I leave is in line 623 that should change the table numbering (table 1 should be table 4).

In Figure 3 a and b (106 DAT), Figure 4 (106 DAT), Table 1 and 2 and Figure 7 instead of putting the letter 'a' in all the data why don't you remove these letters? And put in the caption 'no letters no significant differences' I think it will be easier to visualize, so as it appears it seems that there are differences.

The conclusion could also be improved as it only presents results obtained and not specific conclusions.

In line 716 '2021' must be in bold.

Reviewer 3 Report

Comments and Suggestions for Authors

Topic of paper is very interesting and actual. Generally, I would prefer more-year (2 or 3) results. On the other hand, I accept that it was done in greenhouse, thus effect of different climate was limited. However, I recommend to realise experiments with more cultivars in the future because effect of genotype is very strong to final results.

General comments

- I would prefer to use more precise (shorter) abbreviation for description of treatments. Sometimes, it affect more complicated for reading of results etc.

Material and methods

- line 540: temperature seems to be very high - probably it had impact on the yield (from aspect of pollination) and quality of fruits and then particular qualitative parameters: it could be helpful to fill temperature in shorter time periods (e.g. 10-days intervals, halfs of month etc.)

- line 529-540 - 4.1 Experimental conditions and plant material:

a) fill information about irrigation and protection against pests (temperature was high, air humidity low - ideal conditions for pest occurence which can affect fruit quality)

b) there is nothing about basic fertilisation (NPK) for tomato plants. Please, add it

- line 547-550: add producer+country of nanochitosan and KI

- line 571-577: there is not clear how many fruits were used for preparation of average sample used to analyses

Results

- line 120: I do not know why just total proteins were analysed. Generally, their content in vegetables is not high, compared to other foods. Thus, if it has connection to other results or it has another meaning, fill it, please.

- line 138: I would prefer to mention importance of total enzymatic activity for the study.

Discussion

I recommend to add information about iodine limit in connection to risk intake by people. Then, recommended daily amount (RDA) of iodine can be mentioned and compared to your results, e.g. which amount of tomato fruits should be consumed for fulfilment of RDA.

Conclusion

- in my opinion, just abbreviations should not be used in this way

- add importance/utilisation of your results for growing practice

Final comment

Paper is well-prepared. In the area of biostimulant or fertilisation generally, yield should be primarily determined and then economical effect. Thus, practical importance of this study for growing practice is decreased because yield was not analysed in paper. From practical view, quality is on the secondary place. 

Round 2

Reviewer 1 Report

Comments and Suggestions for Authors

The authors address all the issues mentioned in the review. 

Author Response

The authors address all the issues mentioned in the review. 

  1. Thank you for your valuable time and comments.

Reviewer 3 Report

Comments and Suggestions for Authors

I appreciate that authors have been worked on article and accept most of recommendations. 

line 119 (results) or 343 (Discussion): in first review, I have this note: "I do not know why just total proteins were analysed. Generally, their content in vegetables is not high, compared to other foods. Thus, if it has connection to other results or it has another meaning, fill it, please" - please, fill short information about it as introduction to results focused on protein content or in discussion.

Answer to author note in Final comment

Author reply: Thank you for your final comment. Although this article only includes the enzymatic, biochemical, mineral content and shelf life aspects, a previously published scientific note contains the results of the agronomic variables and the impact generated by the application of treatments. This article is available at: https://doi.org/10.19136/era.a11n2.3623

My reply to this: I understand but in future it should not be divided in this way, I think. Maybe, it could be interesting to have some short information about impact to yield and then noted that also quality can be also affected. (something in this way)

Author Response

I appreciate that authors have been worked on article and accept most of recommendations. 

line 119 (results) or 343 (Discussion): in first review, I have this note: "I do not know why just total proteins were analysed. Generally, their content in vegetables is not high, compared to other foods. Thus, if it has connection to other results or it has another meaning, fill it, please" - please, fill short information about it as introduction to results focused on protein content or in discussion.

R: The comment was resolved by adding the importance of the evaluation of total proteins and their impact on the variables analyzed. In addition, two references were added that speak about the optimal values ​​found in tomato leaves that coincide with the total protein values ​​found in this research. Likewise, a correlation analysis of the total protein content with the variables analyzed was performed and negative correlations were found, so the results are considered consistent with the fact that low stress levels allow a greater accumulation of proteins while higher stress levels translate into an increase in the antioxidant activity of the enzymes SOD, CAD, APX and GPX.

Answer to author note in Final comment

Author reply: Thank you for your final comment. Although this article only includes the enzymatic, biochemical, mineral content and shelf life aspects, a previously published scientific note contains the results of the agronomic variables and the impact generated by the application of treatments. This article is available at: https://doi.org/10.19136/era.a11n2.3623

My reply to this: I understand but in future it should not be divided in this way, I think. Maybe, it could be interesting to have some short information about impact to yield and then noted that also quality can be also affected. (something in this way)

R: Thanks for the final comment, it will be considered for future publications.